# TabularGSM: Understanding the Limitations of LLMs in Tabular Math Reasoning

## Abstract

Mathematical reasoning has long been a key benchmark for evaluating large language models (LLMs). Although substantial progress has been made on math word problems, the need for reasoning over tabular data in real-world applications has been overlooked. For instance, applications such as business intelligence demand not only multi-step numerical reasoning with tables but also robustness to incomplete or inconsistent information. However, comprehensive evaluation in this area is severely limited, constrained by the reliance on manually collected tables that are difficult to scale and the lack of coverage for potential traps encountered in real-world scenarios. To address this problem, we propose AutoT2T, a neuro-symbolic framework that controllably transforms math word problems into scalable and verified tabular reasoning tasks, enabling the evaluation of both accuracy and robustness. Building on this pipeline, we develop TabularGSM, a benchmark comprising three progressively complex subsets and a trap subset, with two complementary evaluation settings. Our study reveals three key observations: (1) Tabular structure makes mathematical reasoning more challenging; (2) The difficulties stem from the joint effects of tabular retrieval and reasoning; (3) Reasoning robustness is another significant issue that needs to be addressed in existing LLMs. In-depth analyses are conducted for each observation to guide future research.

## 1 Introduction

Mathematical reasoning has long been a critical benchmark for evaluating the capabilities of large language models (LLMs). The field has advanced remarkably in recent years (OpenAI, 2023; Guo et al., 2025), with many single-scenario benchmarks now considered largely solved (Hosseini et al., 2014; Patel et al., 2021; Cobbe et al., 2021). This progress has prompted a shift in research focus toward real-world applications, particularly reasoning over semi-structured data like tables (Lu et al., 2022). Unlike plain text, tables present information in a highly structured and organized format, making them indispensable in domains such as business intelligence (Zhang et al., 2024) and financial forecasting (Zhu et al., 2021).

Nevertheless, real-world table reasoning scenarios present significant challenges for LLMs. For example, in the financial sector, the need to process large-scale tables continues to grow with the increasing volume and complexity of data, alongside stricter requirements for reliability and security (Bradley et al., 2024; Zavitsanos et al., 2024). In quarterly financial reports, models are expected not only to perform cross-column computations on numerous metrics like revenue, profit, and liabilities but also to verify numerical consistency (e.g., ensuring total assets equal the sum of liabilities and equity). Failure to properly interpret the data or detect inconsistencies can lead to severe consequences in downstream applications like investment decisions and risk assessment (Cerchiello & Giudici, 2016).

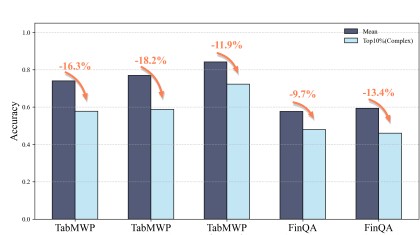

Figure 1: Model performance comparison between average questions and top10% complex questions.

Despite prior works (Zhu et al., 2021; Chen et al., 2021; Lu et al., 2022) addressing some aspects of tabular mathematical reasoning, these efforts have been limited in table scale and primarily focused

on accuracy of perfectly crafted problems. Specifically, existing tabular benchmarks largely rely on manual annotation and collection, making it difficult to scale the datasets effectively. As a result, these benchmarks fail to explore the limits of LLMs' reasoning capabilities on more complex tables, where the models perform worse (as shown in Fig. 1). Then, current benchmarks have not adequately assessed the robustness of tabular mathematical reasoning, overlooking the risk of LLMs providing hallucinated answers when faced with incomplete and inconsistent data. Therefore, to systematically assess model capabilities across multiple dimensions, a comprehensive benchmark is crucial. In this context, there is an urgent need for a more complete and systematic evaluation framework to thoroughly explore and challenge the boundaries of existing models.

To address the above limitations, we propose an **Auto**matic **T**ext-**t**o-**T**able generation framework, AUTOT2T. It is a neuro-symbolic pipeline that converts math word problems into scalable and verified tabular reasoning tasks without human annotation, enabling the evaluation of both accuracy and robustness. To facilitate standardized evaluation and fair comparison, we construct a comprehensive tabular math reasoning benchmark **TabularGSM** based on AUTOT2T. It includes three progressively difficult subsets (*Easy, Medium, Hard*) as well as a *Trap* subset aimed at evaluating the robustness of models in the face of incomplete or inconsistent tabular data, covering both table complexity and robustness dimensions. Based on this, we conduct systematic experiments and analyses on 18 open-source and proprietary models, arriving at the following key observations.

1. **Tabular structure makes mathematical reasoning more challenging** (Sec 5.1). We observe that nearly all models suffer significant performance drops on TabularGSM, with proprietary reasoning models showing up to a 60% decline on complex tables. We further analyze the preferences of different types of models for various table formats.

2. **The difficulties stem from the joint effects of tabular retrieval and reasoning**(Sec 5.2). Our findings indicate that reasoning and retrieval difficulty jointly constrain performance. The difficulty of reasoning in tables is significantly higher than pure retrieval, with the performance gap between math reasoning and direct retrieval in tables exceeding 20% on average.

3. **Reasoning robustness is another significant issue that needs to be addressed in existing LLMs** (Sec 5.3). We find that most models exhibit clear robustness weaknesses, often falling into hallucination or overconfidence. Such traps introduce an additional step of reasoning challenge, with contradiction-based traps hidden in reasoning chains being the hardest to detect.

Overall, we provide a systematic and in-depth analysis of tabular mathematical reasoning, revealing that the coupling between reasoning and retrieval forms a core bottleneck in model performance. This study represents a novel attempt at multimodal reasoning over structured data, and we discuss initial directions for addressing these challenges.

## 2 RELATED WORK

**Math Reasoning.** Mathematical reasoning serves as a key benchmark for evaluating the capabilities of large language models (LLMs) due to its verifiable nature. Early progress was made on elementary-level math problems using datasets such as GSM8K (Cobbe et al., 2021), Multi-Arith (Koncel-Kedziorski et al., 2016), and SVAMP (Patel et al., 2021), where methods like in-context learning (Wei et al., 2022; Gao et al., 2023), supervised fine-tuning (Li et al., 2024b), and reinforcement learning (Guo et al., 2025) demonstrated strong performance. Since then, researchers have questioned the accuracy of current assessments of large models' mathematical reasoning, exploring approaches such as neural-symbolic methods (Mirzadeh et al., 2024). A growing area of interest is the robustness of mathematical reasoning (Zhou et al., 2024; Shi et al., 2023), specifically, whether models can refrain from generating hallucinations when faced with incomplete or logically deceptive prompts (Tian et al., 2024; Zhao et al., 2024).

**Table Question Answering.** Table Question Answering(Table QA) has significant practical applications across various domains, including financial statement analysis (Chen et al., 2021) and medical diagnosis (Hasny et al., 2025). The field has advanced considerably with the development of high-quality datasets, beginning with the pioneering work of Pasupat et al. (Pasupat & Liang, 2015), who constructed the WikiTableQuestions (WTQ) dataset using Wikipedia tables. Subsequent research shifted to more complex QA tasks requiring reasoning capabilities, exemplified by datasets such as ToTTo (Parikh et al., 2020)(focused on answer generation) and OTTQA (Chen et al., 2020)

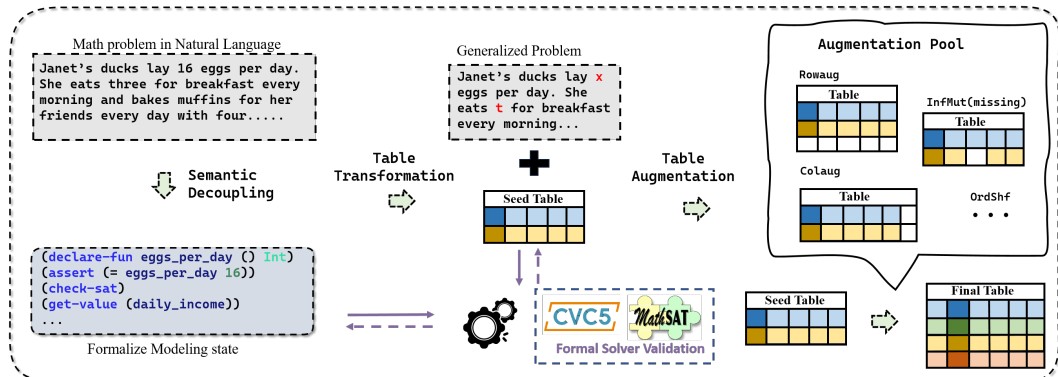

Figure 2: An overview of AUTOT2T pipeline.

(emphasizing cross-table reasoning). More recently, FinQA (Chen et al., 2021) and AiTQA (Katsis et al., 2021) have explored numerical reasoning in tables, while TableBench (Wu et al., 2025) and Text2Analysis (He et al., 2024) introduced multimodal approaches incorporating visual elements. However, most existing datasets rely on manual annotation, lacking an automated pipeline for scalable data processing.

## 3 AUTOMATED TEXT TO TABLE

The construction process of the AUTOT2T (Automated Text to Table) pipeline for converting math word problems into math tabular problems consists of three main stages: semantic decoupling, table transformation, and table augmentation. An overview of AUTOT2T is shown in Fig 2.

### 3.1 SEMANTIC DECOUPLING

Firstly, our objective is to semantically decouple the text of the math word problems and extract key elements that can be structurally represented. Inspired by previous work (Li et al., 2024a), we seek to decompose and verify mathematical problems through formal language modeling. Specifically, we use SMT-Lib (Barrett et al., 2010) as the formal language, and a math problem can be structured into the following forms:

$$
\begin{aligned}
\textit{Goal} \quad & g := \text{solve } f(v) \\
\textit{Constraints} \quad & c := e_1(v) \bowtie e_2(v), \quad \bowtie \in \{\geq, \leq, >, <, =, \neq\} \\
\textit{Expressions} \quad & e := h \mid e_1 \oplus e_2, \quad \oplus \in \{+, -, \times, \div\} \\
\textit{Domains} \quad & \mathcal{D} := \mathbb{N} \mid \mathbb{N}^+ \mid \mathbb{R}
\end{aligned}
$$

In this formulation, $v$ denotes a variable in the problem, $c$ denotes a constraint involving variables, $e$ denotes an expression, $h$ denotes a constant and $f$ denotes the objective function to be solved. For problem $p$, we denote the modeling state as $\mathcal{S} = (\mathcal{V}, \mathcal{C})$ where $\mathcal{V}$ is the set of variables $v$ and $\mathcal{C}$ is the set of constraints $c$. We prompt LLM to perform formal language modeling, breaking down the natural language text $p$ into components of $\mathcal{S}$.

$$(\mathcal{V}, \mathcal{C}) = LLM_{sd}(p) \tag{1}$$

We then invoke a formal solver $\Phi$ (such as Z3 (de Moura & Bjørner, 2008), CVC5 (Barbosa et al., 2022)) to perform symbolic computation and verify whether the derived solution matches that of the original problem. If it is not satisfied, we will re-execute this step. The verification through formal tools ensures the correctness of our conversion.

$$R_{valid} = \Phi(\mathcal{V}, \mathcal{C}) \tag{2}$$

```
Question: Judy is a experienced teacher. She teaches x dance classes, every day, on
the weekdays and y classes on Saturday. If each class has z students and she
charges $w per student, how much money does she make in 1 week?
```

| Name | weekday_cl asses | saturday_c lasses | weekday s | Heart Rate | students_p er_class | charge_per_ student | Body Weight |
|------|------|------|------|------|------|------|------|
| Charlotte | 4 | 1 | 5 | 76 | 27 | 29.1 | 60 |
| Judy 💡 | 5 | 8 | 5 | 83 | 15 | 15.0 | 55 |
| Owen | 9 | 5 | 5 | 90 | 28 | 40.1 | 47 |
| Jackson | 5 | 4 | 5 | 77 | 49 | 32.2 | 74 |

target information     irrelevant columns     ground-truth: 7425

```
…… The total number of weekday
classes in a week is: 5 days* 5
classes/day = 22.5 ……
# Calculation error
```
```
…… each category.We know that Judy
teaches 5 weekday classes and 6
Saturday classes ……
# Retrieval error
```

TableGPT 7B                                    Llama3.1 8B

Figure 3: Illustrative cases in TabularGSM and corresponding model responses

## 3.2 TABLE TRANSFORMATION

After obtaining the formal modeling state, the next goal is to transform semantically decoupled components into a structured tabular representation. Through semantic analysis, we focus on a single entity (e.g., a person) by creating a `name` key and designating it as the table's primary key. Next, we prompt the LLM to convert the variable set $\mathcal{V}$ and their corresponding assigned constraints $\mathcal{C}_a$ into corresponding column fields, while removing the data portion from the original text expression $p$. The LLM output consists of a blurred text description $\hat{p}$ and a *2-row × n-column* seed table $t_{\text{seed}}$, where the first row represents the column headers, the first column represents the primary key entity (`name`), and the remaining cells are filled with initial variables and their assigned values.

$$(\hat{p}, t_{seed}) = LLM_{tt}(p, \mathcal{V}, \mathcal{C}_a) \tag{3}$$

Upon completing the transformation process, we first verify whether a valid entity can be successfully extracted—specifically, whether an entity-centric structure exists (i.e., the presence of a valid `name` field). Subsequently, we reintroduce the variables and their assigned values extracted from the seed table $t_{\text{seed}}$ into the previously constructed formal constraint system $\mathcal{S}$, establishing a new assignment constraint set $\mathcal{C}_{\hat{a}}$, and use the solver $\Phi$ to verify it again.

$$\hat{R}_{valid} = \Phi\left(\mathcal{V}, (\mathcal{C} \setminus \mathcal{C}_a) \cup \mathcal{C}_{\hat{a}}\right) \tag{4}$$

## 3.3 TABLE AUGMENTATION

After obtaining the initial seed table $t_{\text{seed}}$, we construct an augmentation pool $\mathcal{A}$ to expand and modify tables through randomized operations. We provide four types of strategies—*Row Augmentation (RowAug)*, *Column Augmentation (ColAug)*, *Order Shuffling (OrdShf)* and *Information Modification (InfMut)*. The first three strategies increase the table's complexity and retrieval difficulty by adding rows or columns or by shuffling their order, whereas the last strategy introduces traps by altering the target row, making the problem unsolvable. For instance, a missing condition problem removes a required value, while a contradictory condition problem inserts an intermediate variable whose assigned value conflicts with that derived from the known variables. Detailed explanations of these augmentations are provided in Appendix A.2.1.

The user can select a series of augmentation strategies and times based on their own need, and apply them to the previously obtained root table. After each completion, we will get a new table, which will become the input for the next conversion.

$$t_i = \begin{cases} t_{seed} & \text{if } i = 0 \\ Aug_j(t_{i-1}), & Aug_j(\cdot) \in \mathcal{A} \quad \text{if } i > 0 \end{cases} \tag{5}$$

Table 1: Comparation between TabularGSM and existed dataset

| Dataset | Key statistic | | Content Coverage | | | Constrction Process | |
|---|---|---|---|---|---|---|---|
| | Test size | Table cells | Information Retrieval | Math Reasoning | Robustness | Automated Varification | Symbolizable |
| AddSub (Hosseini et al., 2014) | 600 | NA | ✗ | ✓ | ✗ | ✗ | ✗ |
| SVAMP (Patel et al., 2021) | 1000 | NA | ✗ | ✓ | ✗ | ✗ | ✗ |
| GSM8k (Cobbe et al., 2021) | 1438 | NA | ✗ | ✓ | ✗ | ✗ | ✗ |
| PMC (Tian et al., 2024) | 5374 | NA | ✗ | ✓ | ✓ | ✓ | ✗ |
| Tabfact (Chen et al., 2019) | 1695 | 15.1 | ✓ | ✗ | ✗ | ✗ | ✗ |
| FinQA (Chen et al., 2021) | 1147 | 24.5 | ✓ | ✓ | ✗ | ✗ | ✗ |
| TabMWP (Lu et al., 2022) | 7686 | 11.8 | ✓ | ✓ | ✗ | ✗ | ✗ |
| TaT-QA (Zhu et al., 2021) | 669 | 37.6 | ✓ | ✓ | ✗ | ✗ | ✗ |
| TabularGSM | 3391 | 93.5 | ✓ | ✓ | ✓ | ✓ | ✓ |

Through our AUTOT2T pipeline, users can perform customized tabular rewriting of math word problems to automatically generate a large number of diverse problem variants without manual annotation. This solution enables efficient and controllable data generation for specific downstream proprietary domains.

# 4 TABULARGSM BENCHMARK

## 4.1 BENCKMARK DETAILS

Table 2: Key statistics in TabularGSM

| Statistic | Easy | Medium | Hard | Trap |
|---|---|---|---|---|
| Total questions | 797 | 797 | 797 | 1000 |
| Table cells | 41 | 82 | 162 | 90 |
| Table Rows | 4.1 | 4.1 | 8.1 | 4.5 |
| Table Columns | 10 | 20 | 20 | 20 |
| Question Length | 232.2 | 232.2 | 232.2 | 237 |
| RowAug | ✓ | ✓ | ✓ | ✓ |
| OrdShf | ✗ | ✓ | ✓ | ✓ |
| ColAug | ✗ | ✗ | ✓ | ✗ |
| InfMut | ✗ | ✗ | ✗ | ✓ |

To enable standardized evaluation and fair comparison, we construct a predefined benchmark dataset, TabularGSM, through the AU-TOT2T pipeline. Based on the GSM8K Cobbe et al. (2021) test set as the source corpus, TabularGSM consists of four subsets: *Easy*, *Medium*, *Hard*, and *Trap*, comprising a total of 3,391 problems.

**Pure test setting.** For the Pure Setting, we only conceal the information required to solve the problem within the table. The first three subsets each contain 797 examples (with the remaining samples excluded due to failures in table-conversion validation), all derived from the same seed problems. The progression from *Easy* to *Hard* reflects increasing table complexity (in terms of the number of rows or columns), which makes information retrieval more challenging.

**Robust test setting.** We are the first to introduce robustness evaluation in the context of tabular mathematical reasoning, with the primary goal of testing whether models can detect incomplete or inconsistent information in tables and appropriately abstain from answering. Specifically, in *Trap* subdatset, we embed traps into tables, which fall into two categories: **missing-type** (removing essential information from the target line of the table) and **contra-type** (injecting intermediate variables required for the question but designed to conflict with existing information). During evaluation, we observe whether models can accurately identify these traps and abstain from answering, which serves as the core metric. We measure performance by reporting the proportion of ill-defined questions that a model successfully rejects. To avoid overly conservative behavior (e.g., refusing to answer all questions), our robust testing setting mixes well-defined(Medium difficulty) and trap questions in a 1:1 ratio, while explicitly informing the model in the prompt that traps may exist. This setup enables a fair assessment of robustness.

Table 2 summarizes the key statistics of each subset as well as the augmentation strategies employed. And an illustrative case in TabularGSM and corresponding model responses are shown in Fig 3, while additional examples from TabularGSM and visual comparisons with GSM8K are provided in Appendix A.2.

Table 3: Main reults on TabularGSM benchmark

| Dataset | GSM8k | Fmt | Pure test | | | | Robustness test | | | |
|---|---|---|---|---|---|---|---|---|---|---|
| | | | Easy | Medium | Hard | Avg | Well | Contra | Missing | Avg |
| **Open source General Model** | | | | | | | | | | |
| Qwen3 14B | 94.54 | Se | 79.94 | 70.94 | 69.79 | 73.55 | 58.55 | 28.57 | 69.23 | 54.15 |
| | | Md | 77.87 | 70.21 | 61.59 | 69.89 | 55.89 | 28.21 | 67.03 | 51.88 |
| Qwen3 8B | 93.30 | Se | 75.17 | 62.62 | 56.99 | 64.92 | 44.44 | 41.25 | 70.97 | 50.35 |
| | | Md | 73.18 | 55.63 | 47.30 | 58.70 | 36.71 | 34.54 | 68.57 | 44.07 |
| Qwen3 4B | 91.79 | Se | 73.53 | 56.85 | 46.47 | 58.95 | 39.63 | 39.02 | 68.67 | 46.77 |
| | | Md | 71.57 | 52.68 | 41.73 | 55.32 | 39.31 | 32.12 | 69.75 | 45.15 |
| Qwen2.5 14B | 93.40 | Se | 79.90 | 68.06 | 62.34 | 70.10 | 59.40 | 6.40 | 22.40 | 36.90 |
| | | Md | 79.21 | 64.10 | 49.09 | 64.13 | 60.00 | 6.80 | 20.00 | 36.70 |
| Qwen2.5 14B coder | 90.68 | Se | 71.59 | 60.38 | 49.90 | 60.62 | 47.40 | 30.00 | 52.40 | 44.30 |
| | | Md | 72.63 | 57.74 | 45.61 | 58.66 | 47.40 | 23.60 | 51.60 | 42.50 |
| Qwen2.5 7B | 82.86 | Se | 35.56 | 21.36 | 19.39 | 25.43 | 39.20 | 13.60 | 34.00 | 31.50 |
| | | Md | 53.92 | 34.45 | 20.64 | 36.33 | 37.40 | 16.00 | 34.80 | 31.40 |
| Qwen2.5 7B coder | 84.71 | Se | 62.35 | 42.01 | 29.79 | 44.47 | 30.80 | 24.40 | 43.60 | 32.40 |
| | | Md | 64.78 | 42.13 | 23.52 | 43.47 | 33.40 | 20.80 | 34.00 | 30.40 |
| Qwen2.5 3B | 80.28 | Se | 36.37 | 22.71 | 16.94 | 25.34 | 2.20 | 84.00 | 91.20 | 44.90 |
| | | Md | 39.74 | 23.96 | 15.68 | 26.46 | 6.20 | 69.60 | 79.20 | 40.30 |
| LLama3.1 8B | 83.69 | Se | 42.84 | 34.93 | 30.01 | 35.92 | 29.00 | 6.40 | 8.40 | 18.20 |
| | | Md | 48.61 | 33.37 | 32.15 | 38.04 | 29.60 | 10.40 | 9.60 | 19.80 |
| **Open-Source Math Model** | | | | | | | | | | |
| Qwen math 7B | 95.45 | Se | 53.69 | 31.09 | 14.59 | 33.12 | 28.60 | 26.40 | 36.40 | 30.00 |
| | | Md | 53.69 | 30.37 | 14.59 | 32.88 | 27.00 | 20.80 | 48.93 | 30.93 |
| DeepSeek math 7B | 80.13 | Se | 13.93 | 6.24 | 3.96 | 8.04 | 2.60 | 50.40 | 51.20 | 26.70 |
| | | Md | 12.81 | 6.60 | 2.04 | 7.15 | 4.00 | 60.40 | 53.60 | 30.50 |
| **Open-Source Tabular Model** | | | | | | | | | | |
| TableGPT 7B | 24.33 | Se | 30.13 | 18.86 | 12.60 | 20.53 | 26.20 | 26.80 | 44.80 | 31.00 |
| | | Md | 30.60 | 16.44 | 17.64 | 21.56 | 30.60 | 23.20 | 46.40 | 32.70 |
| StructLM 7B | 32.97 | Se | 13.74 | 6.12 | 3.24 | 7.70 | 7.20 | 0 | 0 | 3.60 |
| | | Md | 14.78 | 8.28 | 4.44 | 9.17 | 9.60 | 0 | 0 | 4.80 |
| **Closed-Source API** | | | | | | | | | | |
| DeepSeek v3 | 96.36 | Se | 88.45 | 87.27 | 85.71 | 87.14 | 68.60 | 68.40 | 85.20 | 72.70 |
| | | Md | 88.63 | 87.63 | 85.83 | 87.37 | 68.60 | 68.00 | 82.80 | 72.00 |
| GLM-4-plus | 95.07 | Se | 83.37 | 81.15 | 79.83 | 81.45 | 68.80 | 32.80 | 69.60 | 60.00 |
| | | Md | 84.52 | 81.03 | 78.27 | 81.27 | 71.40 | 27.60 | 65.60 | 59.00 |
| GPT 4 | 94.46 | Se | 83.97 | 82.57 | 77.41 | 81.32 | 66.39 | 22.48 | 74.01 | 57.00 |
| | | Md | 85.54 | 78.42 | 75.23 | 79.73 | 64.25 | 21.11 | 80.20 | 57.80 |

*  More results on TabularGSM are shown in appendix.

## 4.2 Comparison with Existing Benchmarks

As shown in Table 1, TabularGSM differs from existing ones in three key aspects: (1) Tables in TabularGSM are more complex, containing more cells, which makes it harder to retrieve useful information; (2) Compared with previous mathematical reasoning datasets, we jointly evaluate both reasoning and retrieval abilities. Relative to other tabular QA datasets, we place particular emphasis on mathematical reasoning while also assessing whether models can effectively identify traps in problems/tables (e.g., flawed or contradictory conditions), thereby enabling robust and safe reasoning; (3) In terms of construction, unlike prior work that relies heavily on manual annotation, we adopt a neuro-symbolic approach AUTOT2T, which rewrites textual problems into tabular form. This allows us to generate multiple table variants for the same seed problem, achieving efficient and controllable data creation.

## 5 Experiments and Results

In this section, we delve into three key research questions: **RQ1.** Does tabular structure introduce additional challenges for mathematical reasoning? **RQ2.** What factors affect the difficulty of tabular mathematical reasoning? **RQ3.** Can models achieve robust tabular mathematical reasoning?

For the experimental setup, we evaluated four major categories of LLMs within TabularGSM, including open-source general-purpose models, open-source math-specialized models and proprietary APIs. For each setting(*Pure Setting* and *Robust Setting*), we evaluate two widely used approaches

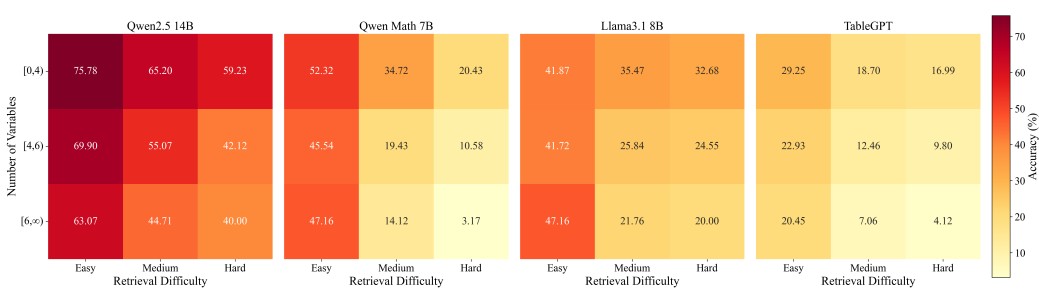

Figure 4: Accuracy heatmap of table complexity and reasoning difficulty in tabular math reasoning

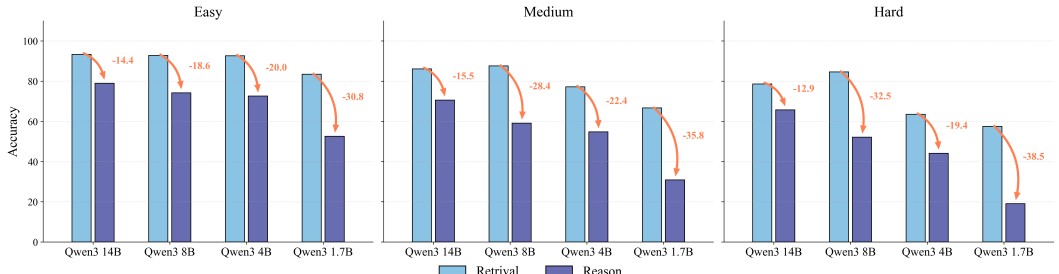

Figure 5: Performance comparison of table reasoning and single-step table retrieval

for organizing tables: Serialized format(Se) and Markdown format(Md). More setup details can be found in Appendix A.3.1.

## 5.1 DOES TABULAR STRUCTURE INTRODUCE ADDITIONAL CHALLENGES FOR MATHEMATICAL REASONING?

**Observation 1: Tabular structure makes mathematical reasoning more challenging.** As shown in Table 3 left side, a consistent and substantial performance drop is observed across all models when transitioning from the original GSM8K dataset to our constructed TabularGSM. Furthermore, this decline becomes more pronounced as table complexity increases. Smaller models—already limited in reasoning capability—experience sharper performance degradation, and domain-specialized models (including both math-specific and table-specific models) also struggle to generalize in this setting. We conducted a manual analysis of selected results and identified four main types of model errors: retrieval omission (failing to recognize the need for retrieval), retrieval mismatch (discrepancy between the ground truth and retrieved table values), expression error (incorrect formulation of the target equation), and numerical calculation error (correct formula but incorrect computation). We found that retrieval mismatch is a primary cause of model errors. A detailed case analysis is provided in Appendix A.4.

**General reasoning models tend to prefer serialized format, while Tabular LLMs show a preference for markdown format.** The results in Table 3 clearly show that the organizational format of the table is indeed important in tabular mathematical reasoning. Overall, the Serialized format yields consistently higher accuracy across most models, as its linearized structure makes information retrieval easier and reduces the difficulty of locating relevant entries during reasoning. In contrast, the Markdown format preserves a more structured layout, which is more natural for humans but introduces additional complexity for general-purpose reasoning models, making retrieval and multi-step reasoning harder. Nevertheless, such structured representations may prove more beneficial for models explicitly trained on table-structured data, such as specialized tabular reasoning LLMs.

## 5.2 WHAT FACTORS AFFECT THE DIFFICULTY OF TABULAR MATHEMATICAL REASONING?

**Observation 2: The difficulties stem from the joint effects of tabular retrieval and reasoning**
We categorize all test data from TabularGSM for each model along two axes: retrieval difficulty (i.e., table complexity, divided into easy, medium, and hard subsets) and reasoning difficulty (the number of variables required to solve the problem). This results in nine categories, and we present the corresponding accuracies as a heatmap in Figure 4. The results show that model performance

Table 4: Accuracy on Medium Dataset: Pure vs. Robust Evaluation

| Model | Pure acc | Robust acc | $\Delta$ |
|---|---|---|---|
| Qwen3 14B | 70.57 | 57.22 | -13.35 |
| Qwen2.5 14B | 66.08 | 59.70 | -6.38 |
| Llama 3.1 8B | 34.15 | 29.30 | -4.85 |
| Qwen math | 30.73 | 23.60 | -7.13 |
| Deepseek-V3 | 87.45 | 68.20 | -19.25 |

Table 5: Comparison between performance on Direct trap and Hidden trap problems

| Model | Type | D-Trap | H-Trap | $\Delta$ |
|---|---|---|---|---|
| Qwen3 8B | Mis | 91.59 | 69.77 | -21.82 |
| | Con | 36.17 | 37.90 | +1.73 |
| Qwen3 4B | Mis | 89.95 | 69.21 | -20.74 |
| | Con | 37.44 | 35.57 | -1.87 |
| Qwen3 1.7B | Mis | 48.51 | 29.87 | -18.64 |
| | Con | 18.36 | 15.63 | -2.73 |

is strongly correlated with both factors, with retrieval difficulty and reasoning difficulty exerting nearly equivalent influence—underscoring the significance of semi-structured (tabular) mathematical reasoning tasks. In addition, we further conduct targeted table analysis by fixing either the number of rows or columns using AUTOT2T (ablations of ColAug and RowAug), as detailed in the Appendix A.3.2. The findings clearly show that increasing both rows and columns substantially degrades tabular reasoning performance.

**The difficulty of reasoning in tables is significantly increased compared to pure retrieval.** Mathematical reasoning in tables can be regarded as a compound task involving both tabular retrieval and mathematical reasoning. We conducted an exploratory experiment comparing end-to-end reasoning with simplified single-step retrieval. Specifically, with a given question like "`Janet's ducks lay x eggs per day...`", we asked "`How many eggs do Janet's ducks lay per day?`", which corresponds directly to the "`eggs-per-day`" key in the table. As shown in Figure 5, models performed markedly better on the single-step retrieval task than on the full reasoning task. This indicates that when explicitly guided on what to retrieve, models can handle retrieval reliably. However, their performance drops sharply once retrieval is embedded in multi-step reasoning, by an average of 20%. This reveals a pronounced performance gap between retrieval and reasoning in tables.

### 5.3 CAN MODELS ACHIEVE ROBUST TABULAR MATHEMATICAL REASONING?

**Observation 3: Reasoning robustness is another significant issue that needs to be addressed in existing LLMs.** As shown in Table 3 right side, the robust setting is difficult for most LLMs, with poor performance on both well-defined and trap instances. Within traps, contra problems are harder than missing ones. Overall, robustness generally correlates with a model's TableQA ability, but exceptions exist. For example, Qwen2.5 3B uses overly conservative refusal strategies, improving robustness scores while reducing accuracy on well-defined questions, revealing a lack of nuanced robustness understanding. Models tend to focus on direct problem-solving and often fail to check whether problem conditions are satisfied, especially when traps appear during reasoning, which can lead to hallucinations. This is a primary source of robustness vulnerabilities.

**Robustness evaluation as a reasoning challenge involving additional identification steps.** In our robust setting, we explicitly inform the model via prompts about the presence of both well-defined and trap problems. This setup effectively raises the bar for reasoning, as the model must first determine whether a problem is solvable before planning the next steps. As shown in Table 4, this mixed setting leads to a noticeable drop in performance on well-defined problems across all types of models, including advanced proprietary APIs. We argue that this need for discriminative capability essentially places higher demands on the model's planning and reasoning abilities, which makes the model think one step further.

**Traps within the reasoning process increase the risk of model hallucinations.** To further discuss the impact of trap types, we conduct experiments with two types of direct traps (D-Trap): Direct Missing (DM) and Direct Contra (DC). Unlike the traps in TabularGSM, these are more explicit and easier for humans to detect. In DM questions, the table lacks the "name" attribute required by the question, so the target person and their information are entirely missing. In DC questions, the table has duplicate columns with conflicting values, which can produce different answers depending on which value is used. Table 5 shows that models detect missing traps more successfully than indirect traps, while performance on contra traps remains poor and variable. These results suggest that traps within the reasoning process are inherently harder for models to detect.

Table 6: Performance changes on TabularGSM of the basic model after AUTOT2T training

| Settings | Easy | | Medium | | Hard | |
|---|---|---|---|---|---|---|
| | Se | Md | Se | Md | Se | Md |
| Qwen2.5-3B-Instrct | 36.37 | 39.74 | 22.71 | 23.96 | 16.94 | 15.68 |
| Qwen2.5-3B-Instrct + finetune | 55.43 | 53.58 | 41.91 | 41.78 | 37.01 | 36.51 |
| Δ | ↑ 19.06 | ↑ 13.84 | ↑ 19.20 | ↑ 17.82 | ↑ 20.07 | ↑ 20.83 |
| Qwen2.5-7B-Instrct | 35.56 | 53.92 | 21.36 | 34.45 | 19.39 | 20.64 |
| Qwen2.5-7B-Instrct + finetune | 62.47 | 62.10 | 48.06 | 51.32 | 37.14 | 36.39 |
| Δ | ↑ 26.91 | ↑ 8.18 | ↑ 26.70 | ↑ 16.87 | ↑ 17.75 | ↑ 15.75 |

## 5.4 MORE DISCUSSION

**TabularGSM provides a representative range of difficulty levels compared to other dataset.** First, the difficulty of our multiple subsets differs from that of previous benchmarks: our dataset spans a wide range of difficulty levels, encompassing those of other typical datasets. Consequently, model performance on the hard subset of TabularGSM is lower than on conventional benchmarks, reflecting the increased challenge (results in Fig 6). Second, our pipeline allows for flexible dataset expansion and symbolic representation, which not only enables systematic exploration of model capabilities but also mitigates the risk of overfitting on test data.

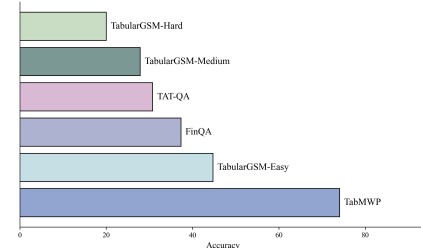

Figure 6: Performance Comparison of Different Datasets on Qwen 2.5-7B

**A flexible data generation framework can provide substantial support to improve tabular math reasoning ability.** Although LLMs have advanced rapidly, they still perform poorly on semi-structured mathematical problems, particularly on TabularGSM. To address this gap, we leverage the flexibility of AUTOT2T to create targeted training data that aligns with tabular reasoning challenges. Using the GSM8K training set as root set, we generated about 6,000 augmented samples and fine-tuned representative models (Qwen2.5-7B and Qwen2.5-3B), As shown in table 6, the fine-tuned model achieves an average performance improvement of approximately 15% in TabularGSM. The benefits also transfer to other tabular reasoning benchmarks such as TAT-QA (Zhu et al., 2021), FinQA (Chen et al., 2021), and TabMWP (Lu et al., 2022), with an average gain of 4%. Improvements are most pronounced on complex tables, highlighting the effectiveness of controlled data generation in enhancing both performance and generalization. Detailed results are provided in Appendix A.3.2.

## 6 CONCLUSION

In this work, we introduce TabularGSM, a comprehensive benchmark for systematically examining the limitations of Tabular math reasoning in LLMs. The benchmark comprises three progressively complex subsets and a trap subset, together enabling precise evaluation of both model performance and robustness. To construct this dataset, we propose a neural-symbolic framework, AUTOT2T, which controllably transforms mathematical word problems into scalable and validated tabular reasoning tasks. We conduct an extensive empirical evaluation of 18 models on TabularGSM, complemented by detailed ablation analyses. By aligning three key observations with three guiding research questions, our study highlights the inherent challenges of tabular mathematical reasoning and offers preliminary insights into potential future research directions.

**Future work and limitations.** Beyond exploring more flexible and practical data generation frameworks, a promising research direction is to decouple retrieval from inference by training large retrieval models tailored for structured data. A limitation of this work is that the evaluation is restricted to large models in the text modality. Future studies could incorporate tabular data in image form and assess the effectiveness of multimodal large language models.

## ETHICS STATEMENT

Our work adheres to the ethical guidelines outlined by the International Conference on Learning Representations (ICLR). The research presented in this article focuses on methodological development and does not involve human subjects, sensitive personal data, or animals. All datasets used are publicly available or synthetically generated, and do not contain sensitive or personally identifiable information. The purpose of this study is to explore the limits of large language models in tabular mathematical reasoning and to provide a standardized benchmark for future research. The experiments were conducted with moderate computational resources and entail a limited environmental footprint. This work is intended solely for academic research.

## REPRODICIBILITY STATEMENT

Our dataset and main evaluation code are publicly available at `https://anonymous.4open.science/r/TabularGSM-2C31/`. The experimental results reported in this paper can be reproduced using the released code.

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

# A APPENDIX

## A.1 USE OF LLMS

This work investigates the boundaries of large language models (LLMs) in tabular mathematical reasoning. In this process, LLMs serve a dual role. On the one hand, they act as both the subject of study and a tool for dataset construction and experimental evaluation, supporting data generation (Sec. 3) and benchmarking representative models (Sec. 4). On the other hand, LLMs are additionally employed to polish the writing and enhance the clarity of English expression.

All key research ideas, theoretical analysis, experimental design, and writing of the main body of the paper were independently completed by the authors. We did not use the large language model to generate the scientific content of the manuscript, nor did we contribute to the formulation of the research hypotheses or the interpretation of the findings. The authors bear full responsibility for the accuracy, originality, and completeness of all content in the paper.

## A.2 DETAILS OF TABLEGSM8K DATASET

To evaluate the reasoning ability of the models on structured data, we construct the TabularGSM benchmark including four subsets: *Easy, Medium, Hard,* and *Robust*. The *Robust* subset contains 50% solvable problems (corresponding to medium difficulty) and 50% unsolvable problems (25% with contradictory conditions + 25% with missing information).

Taking the following problem as an example (Example 1), we use the four categories mentioned in 3.1.3 to generate five tables(Table 7 8 9 10 11) based on the seed row and corresponding generalized problem, so that the table and generalized problem form a question pair as our dataset.

- **Easy:** Apply RowAug (10 times) and Shuffling to the seed row
- **Medium:** Apply RowAug (20 times), Shuffling to the seed row
- **Hard:** Apply RowAug (20 times), Shuffling, and ColAug (4 columns) (adding irrelevant information marked with gray)
- **Robust:** Apply RowAug (20 times), Shuffling, and InfMod(two situations) to the seed row
  - **Contra:** Apply Contradictory Condition Modification, adding row *"eggs_for_sale"*, which is an implicit variable that can be obtained from the formula "*eggs_for_sale = eggs_per_day - eggs_eaten - eggs_for_muffins*". Modify this implicit variable (original value is marked with blue) to create conflicts with existing constraints, making the problem unsolvable.
  - **Missing:** Apply Missing Condition Modification, removing a key data (marked with yellow)(set as null) from seed row.

**Example 1**

**Original Problem:** Janet's ducks lay 16 eggs per day. She eats three for breakfast every morning and bakes muffins for her friends every day with four. She sells the remainder at the farmers' market daily for $2 per fresh duck egg. How much in dollars does she make every day at the farmers' market?

-------------------------------------------------------------------------------------

**Generalized Problem:** Janet's ducks lay x eggs per day. She eats y for breakfast every morning and bakes muffins for her friends every day with z. She sells the remainder at the farmers' market daily for $w per fresh duck egg. How much in dollars does she make every day at the farmers' market?

-------------------------------------------------------------------------------------

**Seed Row:** (red word in table)

| Name | Eggs per Day | Eggs Eaten | Eggs for Muffins | Price per Egg |
|------|--------------|------------|------------------|---------------|
| Janet | 16 | 3 | 4 | 2 |

Table 7: Easy Table

| Name | Eggs per Day | Eggs Eaten | Eggs for Muffins | Price per Egg |
|------|--------------|------------|------------------|---------------|
| Sebastian | 72 | 8 | 16 | 7 |
| Sofia | 73 | 1 | 17 | 7 |
| Elijah | 5 | 4 | 14 | 10 |
| Mia | 73 | 9 | 19 | 7 |
| Ava | 46 | 0 | 7 | 5 |
| Samuel | 3 | 8 | 6 | 7 |
| Logan | 47 | 0 | 9 | 7 |
| Henry | 95 | 9 | 9 | 6 |
| Janet | 16 | 3 | 4 | 2 |
| Ella | 65 | 8 | 15 | 4 |
| Elizabeth | 54 | 0 | 2 | 4 |

Table 8: Medium Table

| Price/Egg | For Muffins | Name | Eaten | Eggs/Day |
|-----------|-------------|------|-------|----------|
| 1 | 18 | Jacob | 1 | 17 |
| 8 | 7 | Sebastian | 0 | 20 |
| 9 | 12 | Lillian | 3 | 59 |
| 6 | 14 | Aiden | 1 | 80 |
| 7 | 20 | Joseph | 2 | 11 |
| 7 | 4 | James | 9 | 20 |
| 9 | 16 | Grace | 4 | 36 |
| 10 | 17 | Mia | 6 | 90 |
| 8 | 8 | Oliver | 3 | 43 |
| 6 | 13 | Charlotte | 0 | 18 |
| 9 | 3 | Mia | 3 | 79 |
| 10 | 15 | Mason | 1 | 34 |
| 1 | 1 | Jacob | 9 | 85 |
| 4 | 8 | Lucas | 8 | 95 |
| 3 | 13 | Liam | 8 | 56 |
| 10 | 5 | James | 8 | 84 |

| Price/Egg | For Muffins | Name | Eaten | Eggs/Day |
|---|---|---|---|---|
| 1 | 0 | Oliver | 5 | 47 |
| 6 | 19 | Eleanor | 5 | 40 |
| 3 | 10 | Victoria | 9 | 68 |
| 5 | 5 | Samuel | 5 | 16 |
| 2 | 4 | Janet | 3 | 16 |

Table 8 – Continued from previous page

Table 9: Hard Table

| Age | Heart Rate | Eggs/Day | Price/Egg | Name | Eaten | For Muffins | Body Temp | Sleep Hours |
|-----|-----------|----------|-----------|------|-------|-------------|-----------|-------------|
| 23 | 81 | 75 | 1 | Emma | 3 | 1 | 38 | 4 |
| 46 | 79 | 81 | 5 | Chloe | 7 | 0 | 38 | 9 |
| 43 | 88 | 10 | 3 | Emma | 2 | 14 | 39 | 6 |
| 42 | 73 | 41 | 9 | Madison | 3 | 11 | 40 | 10 |
| 51 | 87 | 98 | 3 | Eleanor | 0 | 4 | 40 | 9 |
| 50 | 97 | 94 | 6 | Olivia | 1 | 7 | 37 | 10 |
| 63 | 67 | 93 | 3 | Lily | 5 | 16 | 40 | 8 |
| 38 | 70 | 51 | 5 | David | 5 | 11 | 39 | 6 |
| 70 | 87 | 19 | 10 | Isabella | 3 | 17 | 40 | 7 |
| 64 | 99 | 11 | 1 | Avery | 8 | 9 | 38 | 10 |
| 72 | 67 | 81 | 7 | Emily | 1 | 20 | 38 | 4 |
| 57 | 69 | 38 | 6 | Ella | 7 | 16 | 36 | 4 |
| 25 | 62 | 94 | 10 | John | 3 | 11 | 39 | 5 |
| 71 | 91 | 29 | 5 | Camila | 6 | 7 | 38 | 4 |
| 42 | 73 | 62 | 9 | Layla | 7 | 17 | 36 | 8 |
| 62 | 96 | 32 | 7 | Harper | 2 | 19 | 38 | 6 |
| 36 | 78 | 77 | 8 | Olivia | 6 | 3 | 39 | 9 |
| 48 | 85 | 7 | 7 | Aiden | 8 | 10 | 38 | 8 |
| 60 | 82 | 20 | 5 | Joseph | 9 | 19 | 38 | 6 |
| 30 | 94 | 77 | 7 | Logan | 2 | 18 | 40 | 7 |
| 25 | 72 | 16 | 2 | Janet | 3 | 4 | 39 | 5 |

Table 10: Table with Contradictory Conditions

| eggs_per_day | eggs_for_sale (real_eggs_for_sale) | eggs_eaten | name | eggs_for_muffins | price_per_egg |
|--------------|-----------------------------------|------------|------|------------------|---------------|
| 65 | 10 (38) | 7 | Noah | 20 | 4 |
| 87 | 9 (65) | 5 | Wyatt | 17 | 1 |
| 95 | 13 (83) | 0 | Jayden | 12 | 1 |
| 47 | 13 (27) | 8 | Lucas | 12 | 4 |
| 34 | 15 (18) | 9 | Ethan | 7 | 7 |
| 72 | 13 (53) | 7 | Liam | 12 | 2 |
| 79 | 8 (53) | 10 | Sofia | 16 | 4 |
| 12 | 7 (-1) | 5 | Lily | 8 | 9 |
| 58 | 13 (45) | 5 | Sophia | 8 | 1 |
| 31 | 12 (28) | 0 | Jayden | 3 | 5 |
| 90 | 12 (78) | 10 | Ava | 2 | 3 |
| 86 | 16 (58) | 8 | Sophia | 20 | 10 |
| 45 | 14 (42) | 1 | Amelia | 2 | 8 |
| 44 | 16 (37) | 7 | Victoria | 0 | 10 |
| 84 | 10 (64) | 7 | Mason | 13 | 9 |
| 16 | 12 (7) | 3 | Janet | 4 | 2 |
| 74 | 7 (60) | 7 | Oliver | 7 | 10 |
| 43 | 15 (31) | 3 | Aiden | 9 | 8 |
| 82 | 16 (70) | 7 | Michael | 5 | 5 |
| 57 | 16 (45) | 0 | Riley | 12 | 7 |
| 52 | 12 (31) | 5 | Henry | 16 | 8 |

Table 11: Table with Missing Information

| eggs_per_day | eggs_eaten | name | price_per_egg | eggs_for_muffins |
|---|---|---|---|---|
| 66 | 2 | Riley | 8 | 10 |
| 97 | 2 | Hannah | 3 | 0 |
| 70 | 3 | Olivia | 3 | 20 |
| 51 | 8 | Charlotte | 6 | 9 |
| 79 | 0 | Elizabeth | 2 | 13 |
| 16 | null | Janet | 2 | 4 |
| 14 | 10 | Ava | 8 | 3 |
| 48 | 4 | Ethan | 3 | 14 |
| 73 | 7 | Olivia | 3 | 20 |
| 32 | 0 | Chloe | 3 | 14 |
| 41 | 8 | James | 3 | 0 |
| 1 | 1 | Benjamin | 3 | 4 |
| 8 | 0 | Sophia | 3 | 13 |
| 20 | 6 | Victoria | 8 | 14 |
| 93 | 10 | John | 9 | 8 |
| 62 | 0 | Penelope | 10 | 10 |
| 21 | 2 | Harper | 6 | 5 |
| 17 | 1 | Oliver | 10 | 10 |
| 60 | 3 | John | 4 | 4 |
| 14 | 0 | David | 9 | 3 |
| 76 | 0 | Jayden | 1 | 7 |

### A.2.1 DETAILD EXPLANATION OF AUGMENTATIONS

The detaild explanation of augmentations used in $\mathcal{A}$ are as follows.

- **Row Augmentation(RowAug)**: Select an existing row as a seed and modify its column data to simulate information of different individuals (e.g., changing names, adjusting numerical values). As augmented rows only serve to expand the dataset size without affecting the original problem's solvability, no additional validation of numerical rationality is required.

- **Column Augmentation(ColAug)**: A new column is added to the existing table. Since each row is constructed based on the protagonist of the mathematical problem, column augmentation enriches the description of the entities by adding information such as "height", "blood pressure", and other attributes.

- **Order Shuffling(OrdShf)**: Randomly shuffle row or column sequences to increase the difficulty of data retrieval.

- **Information Modification(InfMod)**: This strategy affects solvability in two ways:
  - **Missing Condition Modification**: Remove one or more key data points from seed rows (set as null), rendering the original problem unsolvable due to insufficient conditions.
  - **Contradictory Condition Modification**: Modify implicit variables (values not explicitly stated but derivable from given conditions) to create conflicts with existing constraints, making the problem unsolvable due to logical contradictions.

Table 12: Performance of fine-tuned models on different tabular reasoning benchmarks

| Setting | TabMWP | | FinQA | | TAT-QA | |
|---|---|---|---|---|---|---|
| | Average | Top10% | Average | Top10% | Average | Top10% |
| Baseline | 74.09 | 74.07 | 57.72 | 48.00 | 37.34 | 30.77 |
| Pure-finetune | 91.60 | 91.90 | 74.80 | 71.19 | 67.38 | 51.28 |
| Mix-finetune | 94.70 | 98.11 | 77.59 | 77.97 | 69.53 | 56.41 |
| Δ | ↑ 3.10 | ↑ 6.21 | ↑ 2.79 | ↑ 6.78 | ↑ 2.15 | ↑ 5.13 |

## A.3 EXPERIMENT DETAILS

### A.3.1 SETUP

**Models.** We evaluated four major categories of LLMs within TabularGSM, including open-source general-purpose models (e.g., the Qwen series (Yang et al., 2024a)(including Qwen 3 and Qwen 2.5) and Llama3 series (Grattafiori et al., 2024)), open-source math-specialized models (DeepSeek-Math (Shao et al., 2024) and Qwen-Math (Yang et al., 2024b)), open-source table-specialized models (TableGPT (Zha et al., 2023) and StructLM (Zhuang et al., 2024)), and proprietary API models (GPT-4 (OpenAI, 2023), DeepSeek-v3 (Liu et al., 2024), and GLM-4-plus (GLM et al., 2024)).

**Setting.** The evaluation setup is divided into two parts: *Pure Setting* and *Robust Setting*. In the Pure Setting, models are required to answer questions solely based on the provided tabular information, and we mainly assess their accuracy on standard tabular reasoning tasks. In the Robust Setting, trap problems and medium problems are mixed at a 1:1 ratio to form the test set. At the same time, models are explicitly informed that some problems may be unsolvable. If a model determines that a question cannot be answered based on the given information, it is instructed to output `Unsolvable`.

**Formats.** We evaluate two widely used ways of organizing tables: *serialized format* and *Markdown format*. In the serialized format, each table row is converted into key-value pairs (e.g., `"name: Janet, Eggs_per_day: 16, Eggs_eat_morning: 4..."`). In the Markdown format, the table is presented using standard Markdown syntax, with the first row as column headers and subsequent rows listing values in order, using the `"|"` symbol as the column delimiter. Details prompts can be found in the appendix or the code section.

**Computing Resources.** We use NVIDIA A100 servers as our primary computing platform, along with a few additional machines equipped with RTX 4090 GPUs.

### A.3.2 ADDITIONAL RESULTS

**AUTOT2T-generated data improves performance on other tabular reasoning datasets.** We evaluate on three other tabular math reasoning datasets on Qwen-2.5-7B model, namely TAT-QA (Zhu et al., 2021), FinQA (Chen et al., 2021), and TabMWP (Lu et al., 2022), which primarily test mathematical reasoning over tables. Under the same number of training steps, we further compare two settings: (i) training only on the target dataset's official training set (*Pure-finetune*), and (ii) training on a mixed dataset that combines the target dataset with data generated by AUTOT2T (*Mix-finetune*). As shown in Table 12, the *Mix-finetune* consistently outperforms the *Pure-finetune* setting. The improvement is particularly pronounced on more complex tables, which highlights the versatility and generalization ability of our AUTOT2T across diverse datasets.

**Model performance degrades with increased retrieval difficulty.** First, we want to explore the relationship between performance degradation and table complexity. Through Table 3, we get an initial observation that model performance decreases monotonically as table complexity increases—from easy to hard levels. To further investigate the underlying mechanisms, we conducted a supplementary analysis based on two data augmentation strategies: *ColAug* and *RowAug*. We generated a series of augmented tables by fixing either the number of rows or columns and varying the other, to examine how model performance responds to changes in table structure. As shown in Figure 7, while the inference performance fluctuates as the number of columns (*ColAug*) or rows (*RowAug*) increases, a clear downward trend is evident. We attribute this degradation to the increased

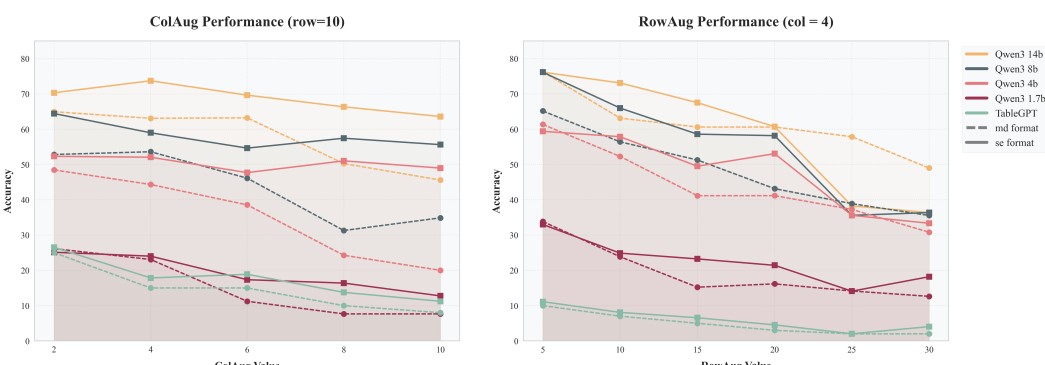

Figure 7: Relationship between model performance changes and table complexity

Table 13: Comparison between performance on Direct trap and Hidden trap problems

| Model | Fmt | Missing | | | Contra | | |
|---|---|---|---|---|---|---|---|
| | | Direct trap | Hidden trap | $\Delta$ | Direct trap | Hidden trap | $\Delta$ |
| Qwen3 14B | Se | 92.30 | 69.23 | -23.07 | 28.68 | 28.57 | -0.11 |
| | Md | 89.18 | 67.03 | -22.15 | 21.47 | 28.21 | 6.74 |
| Qwen3 8B | Se | 93.79 | 70.97 | -22.82 | 40.87 | 41.25 | 0.38 |
| | Md | 89.38 | 68.57 | -20.81 | 31.47 | 34.54 | 3.07 |
| Qwen3 4B | Se | 91.72 | 68.67 | -23.05 | 40.20 | 39.02 | -1.18 |
| | Md | 88.19 | 69.75 | -18.44 | 34.69 | 32.12 | -2.57 |
| Qwen3 1.7B | Se | 50.17 | 37.33 | -12.84 | 20.00 | 16.86 | -3.14 |
| | Md | 46.85 | 22.40 | -24.45 | 16.72 | 14.40 | -2.32 |

presence of irrelevant or distracting information, which raises the difficulty of information retrieval and subsequently impairs the model's reasoning ability.

**Traps within the reasoning process make the model more prone to hallucinations.** Which types of traps are more challenging for the model to detect? To explore this question, we introduce an additional set of experiments by incorporating two types of adversarial scenarios: Direct Missing (DM) and Direct Contra (DC). Unlike the trap questions involved in TabularGSM, these traps are more explicit and thus easier to detect for humans. In the DM questions, the table lacks the "name" attribute required by the question, which means the name of the target person mentioned in the question and their corresponding information are entirely absent from the table. In the DC questions, the table contains two columns with the same header (i.e., duplicate column names) but with conflicting values. These conflicting entries can lead to different answers depending on which value is used. We present our experimental results in Table 13. For missing traps, the model exhibits a significantly higher success rate in identifying direct traps compared to indirect ones. In contrast, performance on contra traps remains consistently poor across models, with some degree of variability. These results indicate that traps embedded within the reasoning process are inherently more difficult to detect.

### A.3.3 DETAILED RESULTS

We provide comprehensive experimental results here that are not given in the main text due to space limitations. Table 14 is the same as Table 3. Table 15 corresponds to Figure 5, which shows the performance comparison of table reasoning and single-step table retrieval. Table 16 17 18 and 19 correspond to Figure 7, which shows the relationship between model performance changes and table complexity.

Table 14: Main reults on TabularGSM benchmark

| Dataset | GSM8k | Fmt | Pure test | | | | Robustness test | | | |
|---|---|---|---|---|---|---|---|---|---|---|
| | | | Easy | Medium | Hard | Avg | Well | Contra | Missing | Avg |
| **Open source General Model** | | | | | | | | | | |
| Qwen3 14B | 94.54 | Se | 79.94 | 70.94 | 69.79 | 73.55 | 58.55 | 28.57 | 69.23 | 54.15 |
| | | Md | 77.87 | 70.21 | 61.59 | 69.89 | 55.89 | 28.21 | 67.03 | 51.88 |
| Qwen3 8B | 93.30 | Se | 75.17 | 62.62 | 56.99 | 64.92 | 44.44 | 41.25 | 70.97 | 50.35 |
| | | Md | 73.18 | 55.63 | 47.30 | 58.70 | 36.71 | 34.54 | 68.57 | 44.07 |
| Qwen3 4B | 91.79 | Se | 73.53 | 56.85 | 46.47 | 58.95 | 39.63 | 39.02 | 68.67 | 46.77 |
| | | Md | 71.57 | 52.68 | 41.73 | 55.32 | 39.31 | 32.12 | 69.75 | 45.15 |
| Qwen3 1.7B | 81.25 | Se | 54.40 | 31.13 | 19.20 | 34.92 | 26.81 | 16.86 | 45.71 | 29.03 |
| | | Md | 50.76 | 30.62 | 18.84 | 33.40 | 33.33 | 14.40 | 37.33 | 29.56 |
| Qwen2.5 14B | 93.40 | Se | 79.90 | 68.06 | 62.34 | 70.10 | 59.40 | 6.40 | 22.40 | 36.90 |
| | | Md | 79.21 | 64.10 | 49.09 | 64.13 | 60.00 | 6.80 | 20.00 | 36.70 |
| Qwen2.5 14B coder | 90.68 | Se | 71.59 | 60.38 | 49.90 | 60.62 | 47.40 | 30.00 | 52.40 | 44.30 |
| | | Md | 72.63 | 57.74 | 45.61 | 58.66 | 47.40 | 23.60 | 51.60 | 42.50 |
| Qwen2.5 7B | 82.86 | Se | 35.56 | 21.36 | 19.39 | 25.43 | 39.20 | 13.60 | 34.00 | 31.50 |
| | | Md | 53.92 | 34.45 | 20.64 | 36.33 | 37.40 | 16.00 | 34.80 | 31.40 |
| Qwen2.5 7B coder | 84.71 | Se | 62.35 | 42.01 | 29.79 | 44.47 | 30.80 | 24.40 | 43.60 | 32.40 |
| | | Md | 64.78 | 42.13 | 23.52 | 43.47 | 33.40 | 20.80 | 34.00 | 30.40 |
| Qwen2.5 3B | 80.28 | Se | 36.37 | 22.71 | 16.94 | 25.34 | 2.20 | 84.00 | 91.20 | 44.90 |
| | | Md | 39.74 | 23.96 | 15.68 | 26.46 | 6.20 | 69.60 | 79.20 | 40.30 |
| LLama3.1 8B | 83.69 | Se | 42.84 | 34.93 | 30.01 | 35.92 | 29.00 | 6.40 | 8.40 | 18.20 |
| | | Md | 48.61 | 33.37 | 32.15 | 38.04 | 29.60 | 10.40 | 9.60 | 19.80 |
| LLama3 8B | 55.34 | Se | 28.92 | 15.22 | 10.63 | 18.25 | 12.80 | 30.80 | 37.20 | 23.40 |
| | | Md | 36.30 | 21.12 | 20.68 | 26.03 | 16.60 | 19.20 | 35.20 | 21.90 |
| **Open-Source Math Model** | | | | | | | | | | |
| Qwen math 7B | 95.45 | Se | 53.69 | 31.09 | 14.59 | 33.12 | 28.60 | 26.40 | 36.40 | 30.00 |
| | | Md | 53.69 | 30.37 | 14.59 | 32.88 | 27.00 | 20.80 | 48.93 | 30.93 |
| DeepSeek math 7B | 80.13 | Se | 13.93 | 6.24 | 3.96 | 8.04 | 2.60 | 50.40 | 51.20 | 26.70 |
| | | Md | 12.81 | 6.60 | 2.04 | 7.15 | 4.00 | 60.40 | 53.60 | 30.50 |
| **Open-Source Tabular Model** | | | | | | | | | | |
| TableGPT 7B | 24.33 | Se | 30.13 | 18.86 | 12.60 | 20.53 | 26.20 | 26.80 | 44.80 | 31.00 |
| | | Md | 30.60 | 16.44 | 17.64 | 21.56 | 30.60 | 23.20 | 46.40 | 32.70 |
| StructLM 7B | 32.97 | Se | 13.74 | 6.12 | 3.24 | 7.70 | 7.20 | 0 | 0 | 3.60 |
| | | Md | 14.78 | 8.28 | 4.44 | 9.17 | 9.60 | 0 | 0 | 4.80 |
| **Closed-Source API** | | | | | | | | | | |
| DeepSeek v3 | 96.36 | Se | 88.45 | 87.27 | 85.71 | 87.14 | 68.60 | 68.40 | 85.20 | 72.70 |
| | | Md | 88.63 | 87.63 | 85.83 | 87.37 | 68.60 | 68.00 | 82.80 | 72.00 |
| GLM-4-plus | 95.07 | Se | 83.37 | 81.15 | 79.83 | 81.45 | 68.80 | 32.80 | 69.60 | 60.00 |
| | | Md | 84.52 | 81.03 | 78.27 | 81.27 | 71.40 | 27.60 | 65.60 | 59.00 |
| GPT 4 | 94.46 | Se | 83.97 | 82.57 | 77.41 | 81.32 | 66.39 | 22.48 | 74.01 | 57.00 |
| | | Md | 85.54 | 78.42 | 75.23 | 79.73 | 64.25 | 21.11 | 80.20 | 57.80 |

Table 15: Model Performance comparison of table reasoning and single-step table retrieval

| Difficulty | Model | se-Retrival | se-Reason | md-Retrival | md-Reason |
|---|---|---|---|---|---|
| Easy | Qwen3 14b | 95.04 | 79.94 | 91.54 | 77.87 |
| | Qwen3 8b | 92.44 | 75.17 | 93.02 | 73.18 |
| | Qwen3 4b | 93.02 | 73.53 | 92.15 | 71.57 |
| | Qwen3 1.7b | 81.10 | 54.40 | 85.71 | 50.76 |
| Medium | Qwen3 14b | 83.28 | 70.94 | 88.85 | 70.21 |
| | Qwen3 8b | 88.26 | 62.62 | 86.80 | 55.63 |
| | Qwen3 4b | 75.00 | 56.85 | 79.41 | 52.68 |
| | Qwen3 1.7b | 63.82 | 31.13 | 69.50 | 30.62 |
| Hard | Qwen3 14b | 75.73 | 69.79 | 81.36 | 61.59 |
| | Qwen3 8b | 86.98 | 56.99 | 82.24 | 47.30 |
| | Qwen3 4b | 60.65 | 46.47 | 66.27 | 41.73 |
| | Qwen3 1.7b | 58.28 | 19.25 | 56.80 | 18.84 |

Table 16: Model Performance Comparison with ColAug (md row=10)

| Model | ColAug2 | ColAug4 | ColAug6 | ColAug8 | ColAug10 |
|-------|---------|---------|---------|---------|----------|
| Qwen3 14b | 64.94 | 63.07 | 63.21 | 50.25 | 45.59 |
| Qwen3 8b | 52.82 | 53.60 | 46.11 | 31.28 | 34.87 |
| Qwen3 4b | 48.45 | 44.32 | 38.54 | 24.26 | 20.00 |
| Qwen3 1.7b | 26.15 | 23.07 | 11.22 | 7.65 | 7.65 |
| LLaMA3 8b | 33.67 | 31.12 | 31.63 | 23.97 | 18.87 |
| TableGPT | 20.40 | 11.73 | 12.24 | 9.18 | 10.20 |

Table 17: Model Performance Comparison with ColAug (se row=10)

| Model | ColAug2 | ColAug4 | ColAug6 | ColAug8 | ColAug10 |
|-------|---------|---------|---------|---------|----------|
| Qwen3 14b | 70.31 | 73.71 | 69.63 | 66.32 | 63.58 |
| Qwen3 8b | 64.43 | 58.97 | 54.63 | 57.43 | 55.61 |
| Qwen3 4b | 52.30 | 52.04 | 47.69 | 51.02 | 48.97 |
| Qwen3 1.7b | 25.12 | 24.01 | 17.34 | 16.38 | 12.75 |
| LLaMA3 8b | 33.67 | 36.22 | 34.18 | 35.71 | 32.14 |
| TableGPT | 26.53 | 17.85 | 18.87 | 13.75 | 11.22 |

Table 18: Model Performance Comparison with RowAug md

| Model | RowAug5 | RowAug10 | RowAug15 | RowAug20 | RowAug25 | RowAug30 |
|-------|---------|----------|----------|----------|----------|----------|
| Qwen3 14b | 76.26 | 63.13 | 60.60 | 60.60 | 57.86 | 48.98 |
| Qwen3 8b | 65.15 | 56.41 | 51.26 | 43.14 | 38.89 | 35.53 |
| Qwen3 4b | 61.34 | 52.28 | 41.14 | 41.16 | 37.24 | 30.80 |
| Qwen3 1.7b | 33.83 | 23.85 | 15.22 | 16.16 | 14.14 | 12.62 |
| LLaMA3 8b | 40.40 | 38.88 | 36.36 | 36.68 | 28.28 | 20.20 |
| TableGPT | 9.18 | 7.65 | 7.65 | 4.59 | 4.59 | 2.04 |

Table 19: Model Performance Comparison with RowAug se

| Model | RowAug5 | RowAug10 | RowAug15 | RowAug20 | RowAug25 | RowAug30 |
|-------|---------|----------|----------|----------|----------|----------|
| Qwen3 14b | 76.14 | 73.09 | 67.51 | 60.71 | 38.25 | 36.36 |
| Qwen3 8b | 76.14 | 65.98 | 58.58 | 58.16 | 35.57 | 36.36 |
| Qwen3 4b | 59.39 | 57.86 | 49.49 | 53.06 | 35.57 | 33.33 |
| Qwen3 1.7b | 32.99 | 24.87 | 23.23 | 21.42 | 14.09 | 18.18 |
| LLaMA3 8b | 43.43 | 40.40 | 33.32 | 29.29 | 25.25 | 26.76 |
| TableGPT | 11.11 | 8.08 | 6.56 | 4.54 | 2.02 | 4.04 |

---

**Semantic Decoupling Prompt**

**"system_prompt":**

> You are an experienced mathematician, and you are familiar with formal languages. I would like you to generate the formal form of a mathematical problem.
>
> You should express all logic in **SMT-LIB syntax**, using **prefix notation**. For example, multiplication should be written as '(* a b)' instead of 'a * b'.
>
> HIGHLIGHT!!!: **All numbers appearing after 'assert' are written as floating point numbers.For example '2' is wrong and it should be replaced with '2.0'.**

------------------------------------------------------------------------

**EXAMPLE INPUT:**

- "problem": "Weng earns \$12 an hour for babysitting. Yesterday, she just did 50 minutes of babysitting. How much did she earn?"

------------------------------------------------------------------------

**EXAMPLE OUTPUT:**

- "problem": "Weng earns \$12 an hour for babysitting. Yesterday, she just did 50 minutes of babysitting. How much did she earn?",

- "formal-problem": "(declare-fun hourly_rate () Int)
  (declare-fun minutes_worked () Real)
  (declare-fun hours_worked () Real)
  (declare-fun earnings () Real)
  (assert (= hourly_rate 12.0))
  (assert (= minute_worked 50.0))
  (assert (= minutes_per_hour 60.0))
  (assert (= hours_worked (/ minutes_per_hour)))
  (assert (= earnings (* hourly_rate hours_worked)))
  (check-sat)
  (get-value (earnings))"

------------------------------------------------------------------------

**"user_prompt:"**

- "problem": {Question}

1188
1189
1190
1191
1192
1193
1194
1195
1196
1197
1198
1199
1200
1201
1202
1203
1204
1205
1206
1207
1208
1209
1210
1211
1212
1213
1214
1215
1216
1217
1218
1219
1220
1221
1222
1223
1224
1225
1226
1227
1228
1229
1230
1231
1232
1233
1234
1235
1236
1237
1238
1239
1240
1241

## Table Transformation Prompt

**"system_prompt":**

The user will provide a problem and its formal representation. You need to convert the **explicitly assigned known data** of the problem into a tabular form.

The table should **only include variables that are directly assigned values in the problem** (e.g., via assertions like `(= variable value)`).

The table should include all variables that appear in the formal definition and their corresponding values: ("Given" or "Calculated").

Please wrap the value of this variable and the method of obtaining it in a list like:
`[5,"Given"]`

**Replace the variables** that appear in the table in the original problem with unknowns to generate a generalized problem (i.e., `table + generalization = original problem`).

Set a **value range for each variable**, ensuring the ranges conform to common sense (they can be fixed values if appropriate).

--------------------------------------------------------------------------------

**EXAMPLE INPUT:**

- "problem": "Weng earns $12 an hour for babysitting. Yesterday, she just did 50 minutes of babysitting. How much did she earn?",

- "formal_problem":
"(declare-fun hourly_rate () Real)
(declare-fun minutes_worked () Int)
(declare-fun hours_worked () Real)
(declare-fun earnings () Real)
(assert (= hourly_rate 12.0))
(assert (= minutes_worked 50))
(assert (= minutes_per_hour 60))
(assert (= hours_worked (/ minutes_worked minutes_per_hour)))
(assert (= earnings (* hourly_rate hours_worked)))
(check-sat)
(get-value (earnings))"

--------------------------------------------------------------------------------

**EXAMPLE OUTPUT:**

- "problem": "Weng earns $12 an hour for babysitting. Yesterday, she just did 50 minutes of babysitting. How much did she earn?",

- "table": [ "name": "Weng",
"hourly_rate": [12,"Given"],
"minutes_worked": [50,"Given"],
"minutes_per_hour": [60,"Given"],
"hours_worked": [0.8333,"Calculated"],
"earnings": [10,"Calculated"] ],

- "generalization": "Weng earns $x an hour for babysitting. Yesterday, she just did t minutes of babysitting. How much did she earn?",

- "value_ranges":
"name": null,
"hourly_rate": "min": 7.25, "max": 100, "unit": "dollars"
"minutes_worked": "min": 10, "max": 1440, "unit": "minutes"
"minutes_per_hour": "min": 60, "max": 60, "unit": "minutes"

--------------------------------------------------------------------------------

**"user_prompt:"**

- "problem": {Question}

- "formal_problem": {Formal_problem}

## A.4 CASE ANALYSIS

### A.4.1 CASE ANALYSIS OF HARD SUBSET

- **Open source General model**: Take **Qwen14b** as an example (Figure 8)
- **Open-Source Math model**: Take **Qwen math 7b** as an example (Figure 9)
- **Open-Source Tabular model**: Take **TableGPT** as an example (Figure 10)
- **Closed source API**: Take **DeepSeek v3** as an example (Figure 11)

### A.4.2 CASE ANALYSIS OF ROBUST SUBSET

- **Open source General model**: Take **Qwen14b** as an example (Figure 12)
- **Open-Source Math model**: Take **Qwen math 7b** as an example (Figure 13)
- **Open-Source Tabular model**: Take **TableGPT** as an example (Figure 14)
- **Closed source API**: Take **DeepSeek v3** as an example (Figure 15)

Figure 8: Hard –Qwen14b

---

**Reason 1: Retrieval Failed**

**Problem:** Eliza's rate per hour for the first x hours she works each week is $y. She also receives an overtime pay of z times her regular hourly rate. If Eliza worked for t hours this week, how much are her earnings for this week?

| OverRatio | OverHour | Name | BodyTemp | Age | RegularRate | Height | HeartRate | RegularHour |
|---|---|---|---|---|---|---|---|---|
| | | | | : | | | | |
| 1.2 | 5 | Eliza | 38 | 62 | 10.0 | 179 | 97 | 40 |
| | | | | : | | | | |

- - - - - - - - - - - - - - - - - - - - - - - - - - - - - - - - - - - - - - - - - - - - - - - - - - - - - - - - - - - -

**Model Response:** *From the provided table, we can find that Eliza's regular hourly rate (y) is $10.0, and her overtime multiplier (z) is 1.5. The number of regular working hours before overtime starts is not directly given but implied to be within the total hours worked as anything beyond that is considered overtime. ......*

---

**Reason 2: Reasoning Failed**

**Problem:** Charlie wants to sell beeswax candles. For every pound of beeswax, he can make x tapered candles. One pound of beeswax and the wicks cost $y in supplies. If he sells each candle for $z each, what is his net profit if he makes and sells t candles?

| HeartRate | BodyTemp | Name | Age | CandlesMade | cost/pound | Height | price | candles/pound |
|---|---|---|---|---|---|---|---|---|
| | | | | : | | | | |
| 83 | 36 | Charlie | 74 | 20 | 10.0 | 199 | 2.0 | 10 |
| | | | | : | | | | |

- - - - - - - - - - - - - - - - - - - - - - - - - - - - - - - - - - - - - - - - - - - - - - - - - - - - - - - - - - - -

**Model Response:** *......we find that Charlie's specific information is: Cost_Per_Pound (y): $10.0 Selling_Price (z):$2.0 Candles_Per_Pound (x): 10. However, there seems to be an inconsistency as the selling price per candle (z) should ideally be higher than the cost of producing one candle to ensure a profit. Typically, the selling price (z) would be higher than the cost per unit, but here, it appears to be lower which would imply a loss rather than profit......*

Figure 9: Hard –Qwen math 7b

---

**Reason 1: Retrieval Error**

**Problem:** Janet's ducks lay x eggs per day. She eats y for breakfast every morning and bakes muffins for her friends every day with z. She sells the remainder at the farmers' market daily for $w per fresh duck egg. How much in dollars does she make every day at the farmers' market?

| Age | HeartRate | eggs/day | price/egg | BodyTemp | eaten | name | for_muffins | SleepHours |
|-----|-----------|----------|-----------|----------|-------|------|-------------|------------|
| | | | | ⋮ | | | | |
| 25 | 72 | 16 | 2 | 39 | 3 | Janet | 4 | 5 |
| | | | | ⋮ | | | | |

- - - - - - - - - - - - - - - - - - - - - - - - - - - - - - - - - - - - - - - - - - - - - - - - - - -

**Model Response:** *......For Janet: Eggs per day: 94 ,Eggs eaten: 3,Eggs for muffins: 4. First, calculate the number of eggs Janet has left after eating and making muffins: 94 - 3 - 4 = 87 ......*

Figure 10: Hard –TableGPT

---

**Reason 1: No Ability to Solve**

**Problem:** Kylar went to the store to buy glasses for his new apartment. One glass costs $x, but every second glass costs only y% of the price. Kylar wants to buy n glasses. How much does he need to pay for them?

| price/glass | glasses | Weight | name | HeartRate | discount | BodyTemp | Height |
|-------------|---------|--------|------|-----------|----------|----------|--------|
| | | | | ⋮ | | | |
| 5.0 | 16 | 98 | Kylar | 98 | 0.6 | 37 | 78 |
| | | | | ⋮ | | | |

- - - - - - - - - - - - - - - - - - - - - - - - - - - - - - - - - - - - - - - - - - - - - - - - - - -

**Model Response:** *This problem is unsolvable.*

---

**Reason 2: Retrieval Failed**

**Problem:** Mishka bought x pairs of shorts, y pairs of pants, and z pairs of shoes. One pair of shorts costs $a. One pair of pants costs $b and one pair of shoes costs $c. How many dollars did Mishka spend on all the clothing items?

| pants | pants price | SleepHours | shorts price | shoes price | shorts | Age | Weight | name | Height | shoes |
|-------|-------------|------------|--------------|-------------|--------|-----|--------|------|--------|-------|
| | | | | | ⋮ | | | | | |
| 3 | 22.5 | 8 | 16.5 | 42.0 | 3 | 21 | 53 | Mishka | 198 | 3 |
| | | | | | ⋮ | | | | | |

- - - - - - - - - - - - - - - - - - - - - - - - - - - - - - - - - - - - - - - - - - - - - - - - - - -

**Model Response:** *......The table provides information about various individuals and their clothing costs, but it does not directly provide the values for x, y, and z (the number of pairs of shorts, pants, and shoes Mishka bought)......*

Figure 11: Hard –DeepSeek v3

---

**Reason 1: Misunderstanding the Question**

**Problem:** John plans to sell all his toys and use the money to buy video games. He has x lego sets and he sells them for $y each. He ends up buying z video games for $w each and has $v left. How many lego sets does he still have?

| name | lego price | Sleep Hours | Body Temp | videogame price | video games | lego sets | Heart Rate | money left | Height |
|------|-----------|-------------|-----------|-----------------|-------------|-----------|------------|-----------|--------|
| | | | | $\vdots$ | | | | | |
| John | 15 | 8 | 40 | 20 | 8 | 13 | 70 | 5 | 159 |
| | | | | $\vdots$ | | | | | |

- - - - - - - - - - - - - - - - - - - - - - - - - - - - - - - - - - - - - - - - - - - - - - - - - - - - - - - - - - - - - -

**Model Response:** ......*Since John plans to sell all his toys, he sells all 13 lego sets. Therefore, he has 0 lego sets left*.

---

**Reason 2: Reasoning Error**

**Problem:** Sasha notices that prices for lumber have gone up x% in the last few months after she bought some lumber. Since she has leftovers, she decides to sell them. She has leftover s small boards that cost her $c each. She also has l large boards she bought for $d each. If she sells them all, how much profit does she make?

| small boards | cost large boards | BodyTemp | name | cost small boards | Age | large boards | Height | Weight | percentage increase |
|--------------|-------------------|----------|------|-------------------|-----|--------------|--------|--------|---------------------|
| | | | | $\vdots$ | | | | | |
| 10 | 16.0 | 39 | Sasha | 10 | 37 | 5 | 150 | 51 | 0.5 |
| | | | | $\vdots$ | | | | | |

- - - - - - - - - - - - - - - - - - - - - - - - - - - - - - - - - - - - - - - - - - - - - - - - - - - - - - - - - - - - - -

**Model Response:** ......*Selling price for small boards: ( c times (1 + x/100) = 10.0 times (1 + 0.5/100) = 10.0 times 1.005 = 10.05*.

Figure 12: Robust –Qwen14b

---

**Reason 1: Retrieval Failed**

**Problem:** Jordan has x children who wear diapers. Each child requires y diaper changes per day. Jordan's wife changes half of the diapers. How many diapers does Jordan change per day?

| children | diaper_changes_per_child | name |
|----------|--------------------------|------|
| ⋮ | | |
| 2 | 5 | Jordan |
| ⋮ | | |

- - - - - - - - - - - - - - - - - - - - - - - - - - - - - - - - - - - - - - - - - - - - - - - - - - - - - - - -

**Model Response:** *........we see various entries for different names and their respective children and diaper change requirements. From the table, the only mention of Jordan is as the name of a child with 2 children needing 5 diaper changes per day. But since the question implies that Jordan (as a parent) has x children, we need to clarify which set of children belong to Jordan as a parent. Since this information isn't clearly specified in the question or table, it's not possible to determine exactly how many children Jordan has as a parent or their specific diaper needs.. ......*

---

**Reason 2: Fail to Identify Contradictions**

**Problem:** Bud makes homemade macaroni and cheese once a week. The pasta costs $x a box, and he spends $y on cheddar cheese and twice that amount for the gruyere cheese. How much money does Bud spend on making macaroni and cheese in one year?

| cost gruyere cheese | weeks per year | total cost week | cost cheddar cheese | cost pasta | name |
|---------------------|----------------|-----------------|---------------------|-----------|------|
| | | ⋮ | | | |
| 10.0 | 52 | 5.0 | 3.0 | 1.0 | Bud |
| | | ⋮ | | | |

- - - - - - - - - - - - - - - - - - - - - - - - - - - - - - - - - - - - - - - - - - - - - - - - - - - - - - - -

**Model Response:** *........The cost of cheddar cheese (y) = 3.0. The cost of gruyere cheese = 10.0. Since the cost of gruyere cheese is twice the cost of cheddar cheese, this aligns with the provided values. The weekly cost for Bud is thus:Total weekly cost = x + y + 2y = 1.0 + 3.0 + 2(3.0) = 1.0 + 3.0 + 6.0 = 10.0......*

Figure 13: Robust –Qwen math 7b

---

**Reason 1: Contradictions Identified but Wrong Answer**

**Problem:** Andrew plans a road trip from New Jersey to Rochester. It takes x days to travel by bus and half as many days to travel by car from New Jersey to Rochester. How many days will he travel if he plans to go to Rochester by bus and return from Rochester by car?

| name | bus_days | car_days |
|------|----------|----------|
| | . | |
| Andrew | 6 | 2 |
| | . | |

--------------------------------------------------------------------------------

**Model Response:** *........From the table, we can see that Andrew has a bus days value of 6 and a car days value of 2. The problem states that it takes half as many days to travel by car as it does to travel by bus. However, the table provides a different value for car days, which is 2. This suggests that the table may have an error or that the problem may have an additional condition that we need to consider. Given the information in the table, we can use the bus days value to calculate the total travel time. The total travel time is the sum of the bus days and the car days. Total travel time = Bus days + Car days = 6 + 2 = 8. Therefore, the total number of days Andrew will travel is 8......*

---

**Reason 2: Missing Identified but Wrong Answer**

**Problem:** Aleksandra went to a restaurant for dinner. She ordered some soup, a bagel, and a piece of cake. The bagel cost $x, and the soup 25% more. The cake is only half of the price of the bagel. How much did Aleksandra need to pay for the dinner she ordered?

| bagel_cost | name |
|------------|------|
| 3.2 | Alexander |
| 15.8 | Aiden |
| 10.7 | Sophia |
| . | |
| null | Aleksandra |
| . | |

--------------------------------------------------------------------------------

**Model Response:** *......we are interested in the cost of the bagel ordered by Aleksandra, which is not specified in the table. However, we can assume that the table is meant to provide a reference for the cost of the bagel, and we can use the average cost of the bagel from the table to find a reasonable estimate.......*

Figure 14: Robust –TableGPT

### Reason 1: Fail to Identify Contradiction

**Problem:** Andrew plans a road trip from New Jersey to Rochester. It takes x days to travel by bus and half as many days to travel by car from New Jersey to Rochester. How many days will he travel if he plans to go to Rochester by bus and return from Rochester by car?

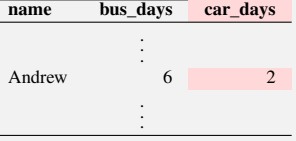

| name | bus_days | car_days |
|------|----------|----------|
| ⋮ | | |
| Andrew | 6 | 2 |
| ⋮ | | |

--------------------------------------------------------------------------

**Model Response:** ........*The table provides the bus and car days for several individuals, including Andrew. From the table: Andrew's bus days: 6 , Andrew's car days: 2. If Andrew plans to go to Rochester by bus and return from Rochester by car, the total number of days he will travel is the sum of his bus days and car days. Total days = Bus days + Car days = 6 + 2 = 8......*

### Reason 2: Regard Missing Info as 0

**Problem:** Samwell owns an appliances store. For this week, one-fourth of their sales are smart TVs, one-eighth are analog TVs, and the rest are OLED TVs. If they were able to sell a total of x TVs, how many OLED TVs were they able to sell?

| total_tvs | name |
|-----------|------|
| 708 | Sebastian |
| 503 | Sophia |
| 935 | Emily |
| ⋮ | |
| null | Aleksandra |
| ⋮ | |

--------------------------------------------------------------------------

**Model Response:** ......*According to the table, Samwell's total TV sales are listed as 0 (None). Since Samwell's total TV sales are 0, we can calculate the number of OLED TVs sold by using the given fractions for smart TVs and analog TVs.One-fourth of the sales are smart TVs: 1/4 times 0 = 0. One-eighth of the sales are analog TVs:1/8 times 0 = 0. The rest of the sales are OLED TVs: 0 - 0 - 0 = 0.......*

Figure 15: Robust –DeepSeek v3

---

**Reason 1: Contradictions Identified but Reasoning Error**

**Problem:**

| first_hour_cost | name | subsequent_hour_cost | total_hours |
|---|---|---|---|
| | | ⋮ | |
| 15 | James | 35 | 4 |
| | | ⋮ | |

---

**Model Response:** *......we find the row corresponding to James: First_Hour_Cost (x): 15, Subsequent_Hour_Cost:35* (**Note: This is not directly used in the calculation since the problem states each subsequent hour is twice the previous hour's cost, not a fixed subsequent cost**) *Total_Hours: 4. First paid hour:15.* *Second paid hour:2 × 15 = 30. Third paid hour:2 × 30 = 60.* *Total Cost = 15+ 3 + 60 = 105...... (The correct answer should be 15+30+30=75)*

---

**Reason 2: Regard Missing Info as 0**

**Problem:** Elaina is holding the final concert in her tour. To celebrate her final concert, she makes the concert twice as long as her usual concerts. At the end of the concert, she also performs a t-minute encore. If the runtime of this final concert is f minutes then how long, in minutes, do her usual concerts run for?

| name | final_concert_length | encore_length |
|---|---|---|
| James | 146 | 22 |
| Ella | 115 | 5 |
| Matthew | 92 | 24 |
| | ⋮ | |
| Elaina | 65 | null |
| | ⋮ | |

---

**Model Response:** *......From the table, we have the following data for Elaina: Final_Concert_Length (f): 65.* *Encore_Length (t): None (which we can interpret as 0 minutes, since no encore is performed)* *According to the problem, the final concert is twice as long as the usual concert, thus x = 65/2 = 32.5.......*