# OpenReview forum: "TabularGSM: Understanding the Limitations of LLMs in Tabular Math Reasoning"
_ICLR.cc/2026/Conference — ICLR 2026 Conference Withdrawn Submission_

### Official Review · Reviewer_qxLi · 2025-10-30

**Soundness:** 2
**Presentation:** 3
**Contribution:** 2
**Rating:** 2
**Confidence:** 4

**Summary:**

This paper proposes AutoT2T to transform math word problems into a robust and large scale tabular-symbolic dataset and provides the benchmark based on this dataset. This paper also shows fine-tuning LLMs on AutoT2T-generated data improves TabularGSM performance by 15+%.

**Strengths:**

- This paper is the first to systematically evaluate robustness in tabular mathematical reasoning—for instance, by testing whether models abstain from answering when data is incomplete or contradictory.
- The tabular dataset used is sufficiently large in scale and more complex than those employed in prior studies.
- The symbolic approach facilitates the creation of diverse instances, effectively avoiding data contamination and enabling a more comprehensive assessment of LLMs’ reasoning capabilities.

**Weaknesses:**

- Limited coverage of multimodal capabilities
This paper only evaluates text-modal tables (e.g., serialized or Markdown formats) while overlooking image-modal tables (e.g., business tables commonly used in real-world scenarios). Additionally, multimodal LLMs such as GPT, Gemini, and Seed are not included in the evaluation, leaving a gap in understanding how visual table structures influence reasoning performance.

- Insufficient justification for the robustness definition
The paper defines robustness as LLMs’ ability to refuse answering questions. However, prompt design and long contextual inputs can also significantly influence LLMs’ responses. Since this is the first formulation of the definition, a clearer justification for its validity and a method to verify its effectiveness (e.g., controlling for prompt/contextual variables) should be provided.

- Inadequate explanatory depth for key results
* Accuracy heatmap (Figure 4): For Llama3.1 8B, the heatmap pattern differs from that of the other three models, but no explanation is provided for this discrepancy.
* Fine-tuning results: While the paper uses small-scale (tiny) models and observes significant performance improvements after fine-tuning, it does not address how these fine-tuned models would perform on other tabular datasets—this limits the generalizability of the findings.

- Lack of substantial innovation in dataset construction
Prior works [1, 2] have already adopted template-based ideas for dataset expansion. Although this paper adds incremental augmentations to the template framework, it still relies on the same core approach, resulting in a lack of substantial innovation.

References
[1] GSM-Symbolic: Understanding the Limitations of Mathematical Reasoning in Large Language Models
[2] Training and Evaluating Language Models with Template-based Data Generation

**Questions:**

- Please see Weaknesses.

---

### Official Review · Reviewer_Utwd · 2025-10-31

**Soundness:** 2
**Presentation:** 2
**Contribution:** 2
**Rating:** 2
**Confidence:** 4

**Summary:**

This paper proposes AUTOT2T, a neuro-symbolic pipeline that converts math word problems into tabular reasoning tasks using SMT-based verification, and builds TabularGSM, a benchmark with Easy/Medium/Hard subsets plus a Trap subset for robustness (missing/contradictory information). The authors evaluate several models and report three core findings: (1) tabularization significantly degrades math performance, (2) retrieval and reasoning jointly drive difficulty, and (3) robustness to ill-posed tables is poor; even strong models hallucinate instead of abstaining. They also show targeted fine-tuning with on ~6k AUTOT2T-generated data  and achieve improved results on TabularGSM (+15%) .

**Strengths:**

Authors in this paper replace manual tabular annotation with a scalable, solver-verified generation process, preserving semantic correctness.

Authors also demonstrate measurable generalization beyond TabularGSM, suggesting AUTOT2T-generated data has reusable inductive value.

The proposed methods show an ~15% average gain on TabularGSM and around +2-3% on TabMWP/FinQA/TAT-QA using 6k GSM8K AUTOT2T samples.

This paper also demonstrates on Controlled RowAug/ColAug/OrdShf augmentations to assess retrieval vs. reasoning difficulty, not just overall accuracy.

Finally, This paper also Introduces a “Trap” subsets (Missing/Contradictory) provides a meaningful safety lens for tabular reasoning.

**Weaknesses:**

Core observations (retrieval bottlenecks, layout sensitivity) reaffirm prior TableQA limitations rather than reveal new phenomena. The headline observations (Observation 1 and 2) about tabular shortcomings (retrieval+reasoning bottlenecks; format effects) are largely known and unsurprising.

Beyond “~6k from GSM8K,” there’s little guidance on how to configure AUTOT2T for data building (augmentation ratios, difficulty mix, domain targeting, per-dataset distribution, release details). Limited detail on AUTOT2T configuration augmentation ratios, difficulty sampling, and reproducibility guidelines.

Built from GSM8K problems, the benchmark may not reflect messy, multi-table, domain-specific spreadsheets and raises potential contamination concerns for API models. GSM8K-derived problems may not reflect multi-table, domain-specific spreadsheet reasoning; contamination risk for API models remains.

Fixing 50% traps inflates the “unsolvable” prior; no sensitivity or calibration analysis over realistic 10–20% rates.  Real-world datasets would rarely have such a high unsolvable rate, so refusal calibration may be misleading. There is no reasoning on how calibrating this percentage affects performance and further fine-tuning.

Evaluation focuses on serialized vs. Markdown; little discussion/experiments with other common encodings (CSV/HTML/JSON), schema randomization (header synonyms, unit perturbations), or stronger tests of row/column invariance beyond OrdShf. Also unclear why Markdown was chosen over other variants. No comparison to semantic parsing approaches (NL→SQL) that are strong on tables.

**Questions:**

Can you please detail the composition of the ~6k AUTOT2T samples: augmentation type proportions (RowAug/ColAug/OrdShf/InfoMut), difficulty mix, and any domain targeting. How should practitioners configure AUTOT2T to cover diversity for a given use case?

For the reported +2–3% gains on TabMWP/FinQA/TAT-QA under mix-finetuning, can you break down which augmentation types/difficulties contributed most? Any learning curves vs. the amount/type of AUTOT2T data?

How do NL→SQL or programmatic (CSV+SQL/Pandas) baselines perform on your splits (including the robust setting)?

Given the GSM8K seed, what steps ensure no leakage for closed-source models, and how representative is TabularGSM of real, messy multi-table spreadsheets? Any experiments with non-GSM seeds?

How sensitive are your robustness results to different trap priors (e.g., 10–20%)? Can you report calibration/ROC-style analyses for “Unsolvable” detection?

Why prioritize Markdown over alternatives? Have you tried CSV/HTML/JSON or header/unit perturbations? How invariant are models to stronger row/column and schema changes beyond OrdShf?

---

### Official Review · Reviewer_HWiC · 2025-11-04

**Soundness:** 2
**Presentation:** 3
**Contribution:** 2
**Rating:** 4
**Confidence:** 4

**Summary:**

This paper introduces a pipeline called AUTOT2T that converts math word problems (MWP) into tabular reasoning tasks. BAsed on this framework, they build TabularGSM, a  benchmark for tabular mathematical reasoning. The goals is to test reasoning complexity and robustness against incomplete or inconsistent data. Experiments on 18 LLMs show that tabular structures make reasoning harder due to the combined difficulty of retrieval and multi-step reasoning. Models frequently produce hallucinated answers when data are missing or contradictory. Fine-tuning models on their generated data improves accuracy and robustness by up to 15% on TabularGSM and 4% on related datasets.

**Strengths:**

Data pipeline
* The use of SMT-Lib and symbolic solvers ensures that the genareted tables are logically correct and consistent.
* Pipeline automation removes the need for manual annotation, allowing large-scale dataset generation.
* Multiple verification steps reduce the likelihood of ill-posed or inconsistent problems.
* The pipeline can be adapted to various domains that require reasoning over tabular data.

Benchmark
* Diff difficulty levels (Easy–Hard–Trap) support more detailed analysis of reasoning complexity and robustness.
* they explicitly integrate “trap” problems (missing and contradictory cases) to test models’ response to ill-posed inputs.
* Experiments provide insights into model behavior under structured data constraints.

Experiments
* They evaluate the benchmarks across multiple model types (general, math, tabular).
* Provde insights by clearly identifying retrieval mismatch as the dominant error cause.
* Give comparison between Serialized and Markdown formats.
* Novel robustness testing with explicit trap detection, addressing a neglected aspect of reasoning evaluation.
* Quantitative analysis (ablations, trap comparisons, etc.) provides further insights into model weaknesses.

**Weaknesses:**

Data pipeline
* Strong reliance on LLM correctness; the semantic parsing and table conversion depend on accurate models, which could introduce errors that propagate through the pipeline.
* The framework is focused on mathematical problems and may not generalize easily to qualitative or textual reasoning tasks.
* The piepline work only for problems that can be mapped to tables with clear entities and attributes, limiting scope.
* Lack of empirical validation for augmentation strategies.
* Can you give more examples for each step of the pipeline? e.g. 10 examples in the appendix where we see how the data is generated/augmented stepwise for these ten examples.

Benchmark
* The benchmark is built solely on GSM8K-derived problems, potentially limiting topic diversity and domain generalization. Why didn;t you consider other benchmarks as well?
* Traps and difficulty tiers are defined synthetically - how well do they reflect real-world inconsistencies?
* Evaluation is based on static performance metrics, did you analyze the generated reasoning traces of models as well?
* Typo in caption for table 3.

Experiments
* Results present accuracy metrics but not a deeper qualitative analysis of reasoning chains.
* Trap design is synthetic and may not reflect real-world data inconsistencies.
* Limited exploration of how reasoning format (e.g., throgh analysis of CoTs) affects performance in tabular settings.
* The don't test if results generalise beyond math reasoning tasks to other structured data domains.
* Since they dont provide a human baseline, its not clear if the low performance of models stems from task/data difficulty or errors in data generation.

**Questions:**

See section weaknesses above.

---

### Note · Authors · 2025-11-19

**Comment:**

Thanks to all the reviewers for their time and valuable suggestions. After careful consideration, I have decided to withdraw the manuscript on behalf of all the authors.

**Withdrawal Confirmation:**

I have read and agree with the venue's withdrawal policy on behalf of myself and my co-authors.